# Adversarial Attacks on Binary Image Recognition Systems

## Abstract

We initiate the study of adversarial attacks on models for binary (i.e. black and white) image classification. Although there has been a great deal of work on attacking models for colored and grayscale images, little is known about attacks on models for binary images. Models trained to classify binary images are used in text recognition applications such as check processing, license plate recognition, invoice processing, and many others. In contrast to colored and grayscale images, the search space of attacks on binary images is extremely restricted and noise cannot be hidden with minor perturbations in each pixel. Thus, the optimization landscape of attacks on binary images introduces new fundamental challenges.

In this paper we introduce a new attack algorithm called SCAR, designed to fool classifiers of binary images. We show that SCAR significantly outperforms existing $L_0$ attacks applied to the binary setting and use it to demonstrate the vulnerability of real-world text recognition systems. SCAR's strong performance in practice contrasts with hardness results that show the existence of worst-case classifiers for binary images that are robust to large perturbations. In many cases, altering a single pixel is sufficient to trick Tesseract, a popular open-source text recognition system, to misclassify a word as a different word in the English dictionary. We also demonstrate the vulnerability of check recognition by fooling commercial check processing systems used by major US banks for mobile deposits. These systems are substantially harder to fool since they classify both the handwritten amounts in digits and letters, independently. Nevertheless, we generalize SCAR to design attacks that fool state-of-the-art check processing systems using unnoticeable perturbations that lead to misclassification of deposit amounts. Consequently, this is a powerful method to perform financial fraud.

## 1 Introduction

In this paper we study adversarial attacks on models designed to classify binary (i.e. black and white) images. Models for binary image classification are heavily used across a variety of applications that include receipt processing, passport recognition, check processing, and license plate recognition, just to name a few. In such applications, the text recognition system typically binarizes the input image (e.g. check processing (Jayadevan et al., 2012), document extraction (Gupta et al., 2007)) and trains a model to classify binary images.

In recent years there has been an overwhelming interest in understanding the vulnerabilities of AI systems. In particular, a great deal of work has designed attacks on image classification models (e.g. (Szegedy et al., 2013; Goodfellow et al., 2014; Moosavi-Dezfooli et al., 2016; Kurakin et al., 2016; Papernot et al., 2016; Madry et al., 2017; Carlini & Wagner, 2017; Chen et al., 2017; Ilyas et al., 2018a;b; Tu et al., 2019; Guo et al., 2019; Li et al., 2019)). Such attacks distort images in a manner that is virtually imperceptible to the human eye and yet cause state-of-the-art models to misclassify these images. Although there has been a great deal of work on attacking image classification models, these attacks are designed for colored and grayscale images. These attacks hide the noise in the distorted images by making minor perturbations in the color values of each pixel.

Somewhat surprisingly, when it comes to binary images, the vulnerability of state-of-the-art models is poorly understood. In contrast to colored and grayscale images, the search space of attacks on binary images is extremely restricted and noise cannot be hidden with minor perturbations of color values in

each pixel. As a result, existing attack algorithms on machine learning systems do not apply to binary inputs. Since binary image classifiers are used in high-stakes decision making and are heavily used in banking and other multi-billion dollar industries, the natural question is:

*Are models for binary image classification used in industry vulnerable to adversarial attacks?*

In this paper we initiate the study of attacks on binary image classifiers. We develop an attack algorithm, called SCAR, designed to fool binary image classifiers. SCAR carefully selects pixels to flip to the opposite color in a query efficient manner, which is a central challenge when attacking black-box models. We first show that SCAR outperforms existing attacks that we apply to the binary setting on multiple models trained over the MNIST and EMNIST datasets, as well as models for handwritten strings and printed word recognition. We then use SCAR to demonstrate the vulnerability of text recognition systems used in industry. We fool commercial check processing systems used by US banks for mobile check deposits. One major challenge in attacking these systems, whose software we licensed from providers, is that there are two independent classifiers, one for the amount written in words and one for the amount written in numbers, that must be fooled with the same wrong amount. Check fraud is a major concern for US banks, accounting for $1.3 billion in losses in 2018 (American Bankers Association, 2020). Since check fraud occurs at large scale, we believe that the vulnerability of check processing systems to adversarial attacks raises a serious concern.

We also show that no attack can obtain reasonable guarantees on the number of pixel inversions needed to cause misclassification as there exist simple classifiers that are provably robust to large perturbations. There exist classifiers for $d$-dimensional binary images such that every class contains some image that requires $\Omega(d)$ pixel inversions ($L_0$ distance) to change the label of that image and such that for every class, a random image in that class requires $\Omega(\sqrt{d})$ pixel inversions in expectation.

**Related work.** The study of adversarial attacks was initiated in the seminal work by Szegedy et al. (2013) that showed that models for image classification are susceptible to minor perturbations in the input. There has since then been a long line of work developing attacks on colored and greyscale images. Most relevant to us are $L_0$ attacks, which iteratively make minor perturbations in carefully chosen pixels to minimize the total number of pixels that have been modified (Papernot et al., 2016; Carlini & Wagner, 2017; Schott et al., 2018; Guo et al., 2019). We compare our attack to two $L_0$ attacks that are applicable in the black-box binary setting (Schott et al., 2018; Guo et al., 2019). Another related area of research focuses on developing attacks that query the model as few times as possible (Chen et al., 2017; Ilyas et al., 2018a;b; Guo et al., 2019; Li et al., 2019; Tu et al., 2019; Al-Dujaili & O'Reilly, 2019). We discuss below why most of these attacks cannot be applied to the binary setting. There has been previous work on attacking OCR systems (Song & Shmatikov, 2018), but the setting deals with grayscale images and white-box access to the model.

Attacks on colored and grayscale images employ continuous optimization techniques and are fundamentally different than attacks on binary images which, due to the binary nature of each pixel, employ combinatorial optimization approaches. Previous work has formulated adversarial attack settings as combinatorial optimization problems, but in drastically different settings. Lei et al. (2018) consider attacks on text classification for tasks such as sentiment analysis and fake news detection, which is a different domain than OCR. Moon et al. (2019) formulate $L_\infty$ attacks on colored image classification as a combinatorial optimization problem where the search space for the change in each pixel is $\{-\varepsilon, \varepsilon\}$ instead of $[-\varepsilon, \varepsilon]$. Finally, we also note that binarization, i.e. transforming colored or grayscale images into black and white images, has been studied as a technique to improve the robustness of models (Schott et al., 2018; Schmidt et al., 2018; Ding et al., 2019).

**Previous attacks are ineffective in the binary setting.** Previous attacks on grayscale (or colored) images are not directly applicable to our setting since they cause small perturbations in pixel values, which is not possible with binary images. One potential approach to use previous attacks is to relax the binary values to be in the grayscale range. However, the issue with this approach is that small changes in the relaxed grayscale domain are lost when rounding the pixel values back to being a valid binary input for the classifier. Another approach is to increase the step size of an attack such that a small change in a grayscale pixel value instead causes a binary pixel value to flip. This approach is most relevant to $L_0$ attacks since they perturb a smaller number of pixels. However, even for the two $L_0$ attacks which can be applied to the binary setting with this approach (Guo et al., 2019; Schott et al., 2018), this results in a large and visible number of pixel inversions, as shown in Section 6.

## 2 PROBLEM FORMULATION

**Binary images and OCR systems.** Binary images $\mathbf{x} \in \{0, 1\}^d$ are $d$-dimensional images such that each pixel is either black or white. An $m$-class classifier $F$ maps $\mathbf{x}$ to a probability distribution $F(\mathbf{x}) \in [0, 1]^m$ where $F(\mathbf{x})_i$ corresponds to the confidence that image $\mathbf{x}$ belongs to class $i$. The predicted label $y$ of $\mathbf{x}$ is the class with the highest confidence, i.e., $y = \arg\max_i F(\mathbf{x})_i$. Optical Character Recognition (OCR) systems convert images of handwritten or printed text to strings of characters. Typically, a preprocessing step of OCR systems is to convert the input to a binary format. To formalize the problem of attacking OCR systems, we consider a classifier $F$ where the labels are strings of characters. Given a binary image $\mathbf{x}$ with label $y$, we wish to produce an adversarial example $\mathbf{x}'$ which is similar to $\mathbf{x}$, but has a predicted label $y' \neq y$. For example, given an image $\mathbf{x}$ of license plate 23FC6A, our goal is to produce a similar image $\mathbf{x}'$ that is recognized as a different license plate number. We measure the similarity of an adversarial image $\mathbf{x}'$ to the original image $\mathbf{x}$ with a perceptibility metric $D_{\mathbf{x}}(\mathbf{x}')$. For binary images, a natural metric is the number of pixels where $\mathbf{x}$ and $\mathbf{x}'$ differ, which corresponds to the $L_0$ distance between the two images. Finding an adversarial example can thus be formulated as the following optimization problem:

$$\min_{\substack{\mathbf{x}' \in \{0,1\}^d \\ \|\mathbf{x} - \mathbf{x}'\|_0 \leq k}} F(\mathbf{x}')_y$$

where $k$ is the maximum dissimilarity tolerated for adversarial image $\mathbf{x}'$. For targeted attacks with target label $y_t$, we instead maximize $F(\mathbf{x}')_{y_t}$. Since there are at least $\binom{d}{k}$ feasible solutions for $\mathbf{x}'$, which is exponential in $k$, this is a computationally hard problem.

**Check processing systems.** A check processing system $F$ accepts as input a binary image $\mathbf{x}$ of a check and outputs confidence scores $F(\mathbf{x})$ which represent the most likely amounts that the check is for. Check processing systems are a special family of OCR systems that consist of two independent models that verify each other. Models $F_C$ and $F_L$ for Courtesy and Legal Amount Recognition (CAR and LAR) classify the amounts written in numbers and in words respectively. If the predicted labels of the two models do not match, the check is flagged. For example, if the CAR and LAR of a valid check read 100 and "one hundred", the values match and the check is processed. The main challenge with attacking checks is to craft an adversarial example $\mathbf{x}'$ with the same target label for both $F_C$ and $F_L$. Returning to the previous example, a successful adversarial check image might have the CAR read 900 and the LAR read "nine hundred". For this targeted attack, the optimization problem is:

$$\max_{\substack{\mathbf{x}' \in \{0,1\}^d, y_t \neq y \\ \|\mathbf{x} - \mathbf{x}'\|_0 \leq k}} F_C(\mathbf{x}')_{y_t} + F_L(\mathbf{x}')_{y_t}$$
$$\text{subject to} \quad y_t = \text{argmax}_i F_C(\mathbf{x}')_i = \text{argmax}_i F_L(\mathbf{x}')_i$$

The attacker first needs to select a target amount $y_t$ different from the true amount $y$, and then attack $F_C$ and $F_L$ such that both misclassify $\mathbf{x}'$ as amount $y_t$. Since check processing systems also flag checks for which the models have low confidence in their predictions, we want to maximize both the probabilities $F_C(\mathbf{x}')_{y_t}$ and $F_L(\mathbf{x}')_{y_t}$. In order to have $\mathbf{x}'$ look as similar to $\mathbf{x}$ as possible, we also limit the number of modified pixels to be at most $k$. Check processing systems are configured such that $F_C$ and $F_L$ only output the probabilities for a limited number of their most probable amounts. This limitation makes the task of selecting a target amount challenging, as aside from the true amount, the most probable amounts for each of $F_C$ and $F_L$ may be disjoint sets.

**Black-box access.** We assume that we do not have any information about the OCR model $F$ and can only observe its outputs, which we formalize with the score-based black-box setting where an attacker only has access to the output probability distributions of a model $F$ over queries $\mathbf{x}'$.

## 3 EXISTENCE OF PROVABLY ROBUST CLASSIFIERS FOR BINARY IMAGES

We first show the existence of binary image classifiers that are provably robust to any attack that modifies a large, bounded, number of pixels. This implies that there is no attack that can obtain reasonable guarantees on the number of pixel inversions needed to cause misclassification. Our first result is that there exists an $m$-class linear classifier $F$ for binary images such that every class contains some image whose predicted label according to $F$ cannot be changed with $o(d)$ pixel flips, i.e., every

class contains at least one image which requires a number of pixel flips that is linear in the number of pixels to be attacked. The analysis, which is in the appendix, uses a probabilistic argument.

**Theorem 1.** *There exists an $m$-class linear classifier $F$ for $d$-dimensional binary images s.t. for all classes $i$, there exists at least one binary image $\mathbf{x}$ in $i$ that is robust to $d/4 - \sqrt{2d \log m}/2$ pixel changes, i.e., for all $\mathbf{x}'$ s.t. $\|\mathbf{x} - \mathbf{x}'\|_0 \leq d/4 - \sqrt{2d \log m}/2$, $\arg\max_j F(\mathbf{x}')_j = i$.*

This robustness result holds for all $m$ classes, but only for the most robust image in each class. We also show the existence of a classifier robust to attacks on an image drawn uniformly at random. There exists a 2-class classifier s.t. for both classes, a uniformly random image in that class requires, in expectation, $\Omega(\sqrt{d})$ pixel flips to be attacked. The analysis relies on anti-concentration bounds.

**Theorem 2.** *There exists a 2-class linear classifier $F$ for $d$-dimensional binary images such that for both classes $i$, a uniformly random binary image $\mathbf{x}$ in that class $i$ is robust to $\sqrt{d}/8$ pixel changes in expectation, i.e. $\mathbb{E}_{\mathbf{x} \sim \mathcal{U}(i)}[\min_{\mathbf{x}':\arg\max_j F(\mathbf{x}')_j \neq i} \|\mathbf{x} - \mathbf{x}'\|_0] \geq \sqrt{d}/8$.*

These hardness results hold for worst-case classifiers. Experimental results in Section 6 show that, in practice, classifiers for binary images are highly vulnerable and that the algorithms that we present next require a small number of pixel flips to cause misclassification.

## 4 ATTACKING BINARY IMAGES

In this section, we present SCAR, our main attack algorithm. We begin by describing a simplified version of SCAR, Algorithm 1, then discuss the issues of hiding noise in binary images and optimizing the number of queries, and finally describe SCAR. At each iteration, Algorithm 1 finds the pixel $p$ in input image $\mathbf{x}$ such that flipping $x_p$ to the opposite color causes the largest decrease in $F(\mathbf{x}')_y$, which is the confidence that this perturbed input $\mathbf{x}'$ is classified as the true label $y$. It flips this pixel and repeats this process until either the perturbed input is classified as label $y' \neq y$ or the maximum $L_0$ distance $k$ with the original image is reached. Because binary images $\mathbf{x}$ are such that $\mathbf{x} \in \{0, 1\}^d$, we implicitly work in $\mathbb{Z}_2^d$. In particular, with $\mathbf{e}_1, \ldots, \mathbf{e}_d$ as the standard basis vectors, $\mathbf{x}' + \mathbf{e}_p$ represents the image $\mathbf{x}'$ with pixel $p$ flipped.

---

**Algorithm 1** A combinatorial attack on OCR systems.

**input** model $F$, image $\mathbf{x}$, label $y$
   $\mathbf{x}' \leftarrow \mathbf{x}$
   **while** $y = \arg\max_i F(\mathbf{x}')_i$ and $\|\mathbf{x}' - \mathbf{x}\|_0 \leq k$ **do**
      $p' \leftarrow \arg\min_p F(\mathbf{x}' + \mathbf{e}_p)_y$
      $\mathbf{x}' \leftarrow \mathbf{x}' + \mathbf{e}_{p'}$
   **return** $\mathbf{x}'$

---

Although the adversarial images produced by Algorithm 1 successfully fool models and have small $L_0$ distance to the original image, it suffers in two aspects: the noise added to the inputs is visible to the human eye, and the required number of queries to the model is large.

**Hiding the noise.** Attacks on images in a binary domain are fundamentally different from attacks on colored or grayscale images. In the latter two cases, the noise is often imperceptible because the change to any individual pixel is small relative to the range of possible colors. Since attacks on binary images can only invert a pixel's color or leave it untouched, noisy pixels are highly visible if their colors contrast with that of their neighboring pixels. This is a shortcoming of Algorithm 1, which results in noise with small $L_0$ distance but that is highly visible (for example, see Figure 1). To address this issue, we impose a new constraint that only allows modifying pixels on the *boundary* of black and white regions in the image. A pixel is on a boundary if it is white and at least one of its eight neighboring pixels is black (or vice-versa). Adversarial examples produced under this constraint have a greater $L_0$ distance to their original images, but the noise is significantly less noticeable.

**Optimizing the number of queries.** An attack may be computationally expensive if it requires many queries to a black-box model. For paid services where a model is hidden behind an API, running attacks can be financially costly as well. Several works have proposed techniques to reduce the number of queries. Many of these are based on gradient estimation (Chen et al., 2017; Tu et al., 2019; Ilyas et al., 2018a;b; Al-Dujaili & O'Reilly, 2019). Recently, several gradient-free black-box

attacks have also been proposed. Li et al. (2019) and Moon et al. (2019) propose two such approaches, but these rely on taking small steps of size $\varepsilon$ in a direction which modifies *all* pixels. SIMBA (Guo et al., 2019), another gradient-free attack, can be extended to the binary setting and is evaluated in the context of binary images in Section 6. We propose two optimization techniques to exploit correlations between pixels both spatially and temporally. We define the *gain* from flipping pixel $p$ at point $\mathbf{x}'$ as the following discrete derivative of $F$ in the direction of $p$:

$$F(\mathbf{x}')_y - F(\mathbf{x}' + \mathbf{e}_p)_y$$

We say a pixel $p$ has *large gain* if this value is larger than a threshold $\tau$.

- **Spatial correlations.** Pixels in the same spatial regions are likely to have similar discrete derivatives (e.g. Figure 4 in appendix). At every iteration, we prioritize evaluating the gains of the eight pixels $N(p)$ neighboring the pixel $p$ which was modified in the previous iteration of the algorithm. If one of these pixels has large gain, then we flip it and proceed to the next iteration.

- **Temporal correlations.** . Pixels with large discrete derivatives at one iteration are likely to also have large discrete derivatives in the next iteration (e.g. Figure 5 in appendix). At each iteration, we first consider pixels that had large gain in the previous iteration. If one of these pixels still produces large gain in the current iteration, we flip it and proceed to the next iteration.

**SCAR.** In order to improve on the number of queries, SCAR (Algorithm 2) prioritizes evaluating the discrete derivatives at pixels which are expected to have large gain according to the spatial and temporal correlations. If one of these pixels has large gain, then it is flipped and the remaining pixels are not evaluated. If none of these pixels have large gain, we then consider all pixels on the boundary $B(\mathbf{x})$ of black and white regions in the image $\mathbf{x}$. In this set, the pixel with the largest gain is flipped regardless of whether it has gain greater than $\tau$. As before, we denote the standard basis vector in the direction of coordinate $i$ with $\mathbf{e}_i$. We keep track of the gain of each pixel with vector $\mathbf{g}$.

---

**Algorithm 2** SCAR, Shaded Combinatorial Attack on Recognition sytems.

---

**input** model $F$, image $\mathbf{x}$, label $y$, threshold $\tau$, budget $k$
    $\mathbf{x}' \leftarrow \mathbf{x}, \mathbf{g} \leftarrow \mathbf{0}$
    **while** $y = \arg\max_i F(\mathbf{x}')_i$ and $\|\mathbf{x}' - \mathbf{x}\|_0 \leq k$ **do**
        **for** $p : g_p \geq \tau$ or $p \in N(p')$ **do**
            $g_p \leftarrow F(\mathbf{x}')_y - F(\mathbf{x}' + \mathbf{e}_p)_y$
        **if** $\max_p g_p < \tau$ **then**
            **for** $p \in B(\mathbf{x}')$ **do**
                $g_p \leftarrow F(\mathbf{x}')_y - F(\mathbf{x}' + \mathbf{e}_p)_y$
        $p' \leftarrow \arg\max_p g_p$
        $\mathbf{x}' \leftarrow \mathbf{x}' + \mathbf{e}_{p'}$
    **return** $\mathbf{x}'$

---

Algorithm 2 is an untargeted attack which finds $\mathbf{x}'$ which is classified as label $y' \neq y$ by $F$. It can easily be modified into a targeted attack with target label $y_t$ by changing the first condition in the while loop from $y = \arg\max_i F(\mathbf{x}')_i$ to $y_t \neq \arg\max_i F(\mathbf{x}')_i$ and by computing the gains $g_p$ as $F(\mathbf{x} + \mathbf{e}_p)_{y_t} - F(\mathbf{x})_{y_t}$ instead of $F(\mathbf{x})_y - F(\mathbf{x} + \mathbf{e}_p)_y$. Even though SCAR performs well in practice, there exists simple classifiers for which any algorithm requires a large number of pixel inversions to find an adversarial example $\mathbf{x}'$, as shown in Section 3.

## 5   SIMULTANEOUS ATTACKS

There are two significant challenges to attacking check processing systems. In the previous section, we discussed the challenge caused by the preprocessing step that binarizes check images (Jayadevan et al., 2012). The second challenge is that check processing systems employ two independent models that verify the output of the other model: $F_C$ and $F_L$ classify the amount written in numbers and in letters respectively. We thus propose an algorithm which tackles the problem of attacking two separate OCR systems simultaneously. A natural approach is to search for a target amount at the intersection of what $F_C$ and $F_L$ determines are probable amounts. However, on unmodified checks, the models are often highly confident of the true amount, and other amounts have extremely small probability. To increase the likelihood of choosing a target amount which will result in an adversarial

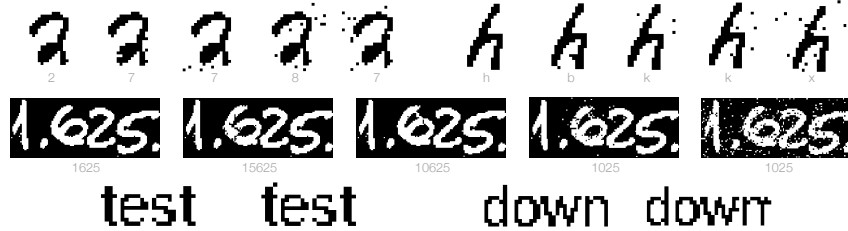

Figure 1: Examples of attacks on a CNN trained over MNIST (top left), a CNN trained over EMNIST (top right), an LSTM for handwritten numbers (center), and Tesseract for typed words (bottom). The images correspond to, from left to right, the original image, the outputs of SCAR, VANILLA-SCAR, POINTWISE, and SIMBA. The predicted labels are in light gray below each image. For Tesseract attacks (bottom), we show the original image and SCAR's output.

example, we first proceed with an untargeted attack on both $F_C$ and $F_L$ using SCAR, which returns image $\mathbf{x}^u$ with reduced confidence in the true amount $y$. Then we choose the target amount $y_t$ to be the amount $i$ with the maximum value $\min(F_C(\mathbf{x}^u)_i, F_L(\mathbf{x}^u)_i)$, since our goal is to attack both $F_C$ and $F_L$. Then we run T-SCAR, which is the targeted version of SCAR, twice to perform targeted attacks on both $F_C$ and $F_L$ over image $\mathbf{x}^u$.

---

**Algorithm 3** The attack on check processing systems.

---

**input** check image $\mathbf{x}$, models $F_C$ and $F_L$, label $y$

   $\mathbf{x}_C, \mathbf{x}_L \leftarrow$ extract CAR and LAR regions of $\mathbf{x}$
   $\mathbf{x}_C^u, \mathbf{x}_L^u \leftarrow \text{SCAR}(F_C, \mathbf{x}_C), \text{SCAR}(F_L, \mathbf{x}_L)$
   $y_t \leftarrow \max_{i \neq y} \min(F_C(\mathbf{x}_C^u)_i, F_L(\mathbf{x}_L^u)_i)$
   $\mathbf{x}_C^t, \mathbf{x}_L^t \leftarrow \text{T-SCAR}(F_C, \mathbf{x}_C^u, y_t), \text{T-SCAR}(F_L, \mathbf{x}_L^u, y_t)$
   $\mathbf{x}^t \leftarrow$ replace CAR, LAR regions of $\mathbf{x}$ with $\mathbf{x}_C^t, \mathbf{x}_L^t$
   **return** $\mathbf{x}^t$

---

## 6 EXPERIMENTS

We demonstrate the effectiveness of SCAR for attacking text recognition systems. We attack, in increasing order of model complexity, standard models for single handwritten character classification (Section 6.2), an LSTM model for handwritten numbers classification (Section 6.3), a widely used open source model for typed (printed) text recognition called Tesseract (Section 6.4), and finally commercial check processing systems used by banks for mobile check deposit (Section 6.5).

### 6.1 EXPERIMENTAL SETUP

**Benchmarks.** We compare four attack algorithms. **SCAR** is Algorithm 2 with threshold $\tau = 0.1$. **VANILLA-SCAR** is Algorithm 1. We compare SCAR to Algorithm 1 to demonstrate the importance of hiding the noise and optimizing the number of queries **SIMBA** is Algorithm 1 in (Guo et al., 2019) with the Cartesian basis and $\varepsilon = 1$. SIMBA is an algorithm for attacking (colored) images in black-box settings using a small number of queries. At every iteration, it samples a direction $\mathbf{q}$ and takes a step towards $\varepsilon\mathbf{q}$ or $-\varepsilon\mathbf{q}$ if one of these improves the objective. In the setting where $\mathbf{q}$ is sampled from the Cartesian basis and $\varepsilon = 1$, SIMBA corresponds to an $L_0$ attack on binary images which iteratively chooses a random pixel and flips it if doing so results in a decrease in the confidence of the true label. **POINTWISE** (Schott et al., 2018) first applies random salt and pepper noise until the image is misclassified. It then greedily returns each modified pixel to its original color if the image remains misclassified. We use the implementation available in Foolbox (Rauber et al., 2017).

**Metrics.** To evaluate the performance of each attack $A$ over a model $F$ and test set $X$, we use three metrics. The **success rate** of $A$ is the fraction of images $\mathbf{x} \in X$ for which the output image $\mathbf{x}' = A(\mathbf{x})$ is adversarial, i.e. the predicted label $y'$ of $\mathbf{x}'$ is different from the true label $y$ of $\mathbf{x}$. We only attack images $\mathbf{x}$ which are initially correctly classified by $F$. We use the $L_0$ **distance** to measure how similar an image $\mathbf{x}' = A(\mathbf{x})$ is to the original image $\mathbf{x}$, which is the number of pixels where $\mathbf{x}$ and $\mathbf{x}'$ differ. The **number of queries** to model $F$ to obtain output image $\mathbf{x}' = A(\mathbf{x})$.

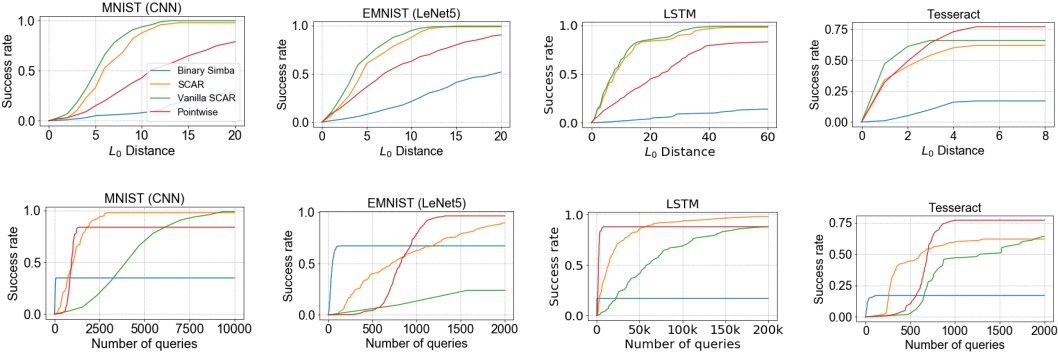

Figure 2: Success rate by $L_0$ distance and number of queries for a CNN model on MNIST, a LeNet5 model on EMNIST, an LSTM model on handwritten numbers, and Tesseract over printed words.

**The distance constraint $k$.** We seek a principled approach to selecting the maximum $L_0$ distance $k$. For an image $\mathbf{x}$ with label $y$, the $L_0$ constraint is $k = \alpha\mathcal{F}(\mathbf{x})/|y|$ where $\mathcal{F}(\mathbf{x})$ counts the number of pixels in the foreground of the image, $\alpha \in [0, 1]$ is a fixed fraction, and $|y|$ represents the number of characters in $y$, e.g. |23FC6A| = 6. In other words, $k$ is a fixed fraction of the average number of pixels per character in $\mathbf{x}$. In our experiments, we set $\alpha = 1/5$.

## 6.2 DIGIT AND CHARACTER RECOGNITION SYSTEMS

For each experiment, we provide further details about the datasets and models in the appendix. We train models over binarized versions of the MNIST digit (LeCun et al., 2010) and EMNIST letter (Cohen et al., 2017) datasets. We binarize each dataset with the map $x \mapsto \lfloor \frac{x}{128} \rfloor$. We additionally preprocess the EMNIST letter dataset to only include lowercase letters. We consider five models: a logistic regression model (LogReg), a 2-layer perceptron (MLP2), a convolutional neural network (CNN), a neural network from (LeCun et al., 1998) (LeNet5), and a support vector machine (SVM). Their Top-1 accuracies are given in the appendix.

We discuss the results of the attacks on the CNN model trained over MNIST and on the LeNet5 model trained over EMNIST. The full results for the remaining 8 models are in the appendix. In Figure 2, we observe that for fixed $L_0$ distances $\kappa \leq k$, VANILLA-SCAR has the largest number of successful attacks with an $L_0$ distance at most $\kappa$ on the CNN model. For example, $80\%$ of the images were successfully attacked by flipping at most 7 of the 784 pixels of an MNIST image. SCAR is very close but requires significantly fewer queries and, as shown in Figure 1 its noise is less visible even though its $L_0$ distance is slightly larger. SIMBA requires very few queries to attack between $40\%$ and $65\%$ of the image, but the attacked images have large $L_0$ distances. The success rate does not increase past $40\%$ and $65\%$ because the noise constraint $k$ is reached. POINTWISE obtains a success rate close to $85\%$ and $98\%$ on the CNN and LeNet5, respectively. The average $L_0$ distance of the images produced by POINTWISE is between SCAR and SIMBA. Overall, SCAR obtains the best number of queries and $L_0$ distance combination. It is the only attack, together with VANILLA-SCAR, which consistently obtains a success rate close to $100\%$ on MNIST and EMNIST models.

## 6.3 LSTM ON HANDWRITTEN NUMBERS

We train an OCR model on the ORAND-CAR-A dataset, part of the HDSRC 2014 competition on handwritten strings (Diem et al., 2014). This dataset consists of 5793 images from real bank checks taken from a Uruguayan bank. Each image contains between 2 and 8 numeric characters. We implement the OCR model described in (Mor & Wolf, 2018). The trained model achieves a precision score of $85.7\%$ on the test set of ORAND-CAR-A, which would have achieved first place in the HDSRC 2014 competition. The results are similar to the attacks on the CNN-MNIST model. SIMBA has less than $20\%$ success rate. POINTWISE obtains a high success rate with a small number of queries, but is outperformed by SCAR and VANILLA-SCAR in terms of $L_0$ distance. Due to the images being high-dimensional ($d \approx 50,000$) and consisting of multiple digits, the reason why SIMBA performs poorly is that the flipped pixels are spread out over the different digits (see Figure 1).

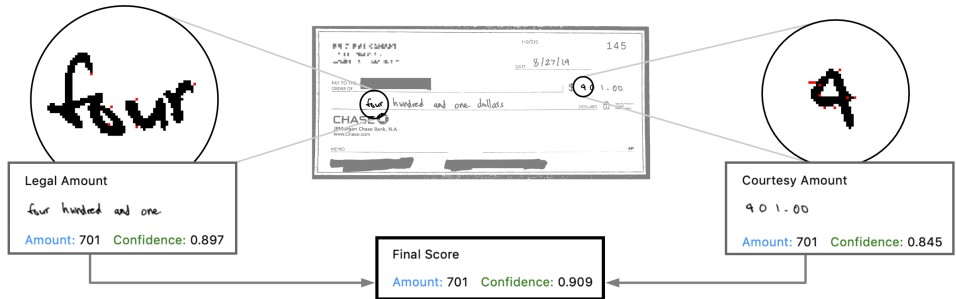

Figure 3: An example of a check for $401 attacked by Algorithm 3 that is misclassified with high confidence as $701 by a check processing system used by US banks.

### 6.4 TESSERACT ON PRINTED WORDS

Tesseract is a popular open-source text recognition system designed for printed text that is sponsored by Google (Smith, 2007). We attacked 100 images of random printed English words of length four (the full list of words, together with the misclassified labels, can be found in the appendix). Tesseract does not recognize any word and reject input images with excess noise. Since the goal is to misclassify images as words with a different meaning, an attack is successful if the adversarial image is classified as a word in the English dictionary. The main result for the attacks on Tesseract is that, surprisingly, for around half of the images, flipping a *single* pixel results in the image being classified as a different word in the English dictionary (see Figure 2). SCAR again produces attacks with $L_0$ distance close to VANILLA-SCAR, but with fewer queries. Unlike the other models, SCAR and VANILLA-SCAR do not reach close to $100\%$ accuracy rate. We hypothesize that this is due to the fact that, unlike digits, not every combination of letters forms a valid label, so many words have an edit distance of multiple characters to get to the closest different label. In these experiments, POINTWISE obtains the highest success rate. In the appendix, we consider SCAR attacking Tesseract on the word "idle" and analyze the spatial and temporal correlations between pixels in that example.

### 6.5 CHECK PROCESSING SYSTEMS

We licensed software from providers of check processing systems to major US banks and applied the attack described in Algorithm 3. This software includes the prediction confidence as part of their output. Naturally, access to these systems is limited and the cost per query is significant. We confirm the findings from the previous experiments that SCAR, which is used as a subroutine by Algorithm 3, is effective in query-limited settings and showcase the vulnerability of OCR systems used in the industry. Check fraud is a major concern for US banks; it caused over $1.3 billion in losses in 2018 (American Bankers Association, 2020). We obtained a $17.1\%$ success rate (19 out of 111 checks) when attacking check processing systems used by banks for mobile check deposits. As previously mentioned, a check is successfully attacked when both amounts on the check are misclassified as the same wrong amount (see Figure 3). Since check fraud occurs at large scale, we believe that this vulnerability raises serious concerns.[1]

| Classifier | Queries | $L_0$ distance |
|------------|---------|----------------|
| CAR ($F_C$) | 1615 | 11.77 |
| LAR ($F_L$) | 8757 | 14.85 |

We say that a check is misclassified with high confidence if the amounts written in number and words are each classified with confidence at least $50\%$ for the wrong label. We obtained high confidence misclassification for $76.5\%$ of the checks successfully attacked. In Figure 3, we show the output of a check for $401 that has both amounts classified as 701 with confidence at least $80\%$. On average, over the checks for which we obtained high confidence misclassification, Algorithm 3 flipped 11.77 and 14.85 pixels and made 1615 and 8757 queries for the amounts in numbers and words respectively. The checks are high resolution, with widths of size 1000. Additional examples of checks misclassified with high confidence can be found in the appendix.

---

[1]Regarding physical realizability: instead of printing an adversarial check in high resolution, an attacker can redirect the camera input of a mobile phone to arbitrary image files, which avoids printing and taking a picture of an adversarial check. This hacking of the camera input is easy to perform on Android.

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

## A   MISSING ANALYSIS FROM SECTION 3

**Theorem 1.** *There exists an $m$-class linear classifier $F$ for $d$-dimensional binary images s.t. for all classes $i$, there exists at least one binary image $\mathbf{x}$ in $i$ that is robust to $d/4 - \sqrt{2d \log m}/2$ pixel changes, i.e., for all $\mathbf{x}'$ s.t. $\|\mathbf{x} - \mathbf{x}'\|_0 \leq d/4 - \sqrt{2d \log m}/2$, $\arg\max_j F(\mathbf{x}')_j = i$.*

*Proof.* In this proof, we assume that binary images have pixel values in $\{-1, 1\}$ instead of $\{0, 1\}$. We consider a linear classifier $F_{\mathbf{w}_1^\star, \dots, \mathbf{w}_m^\star}$ such that the predicted label $y$ of a binary image $\mathbf{x}$ is $y = \arg\max_i \mathbf{x}^\mathsf{T} \mathbf{w}_i^\star$.

We wish to show the existence of weight vectors $\mathbf{w}_1^\star, \dots, \mathbf{w}^{\star m}$ that all have large pairwise $L_0$ distance. This is closely related to error-correction codes in coding theory which, in order to detect and reconstruct a noisy code, also aims to construct binary codes with large pairwise distance.

We do this using the probabilistic method. Consider $m$ binary weight vectors $\mathbf{w}_1, \dots, \mathbf{w}_m$ chosen uniformly at random, and independently, among all $d$-dimensional binary vectors $\mathbf{w} \in \{-1, 1\}^d$. By the Chernoff bound, for all $i, j \in [m]$, we have that for $0 < \delta < 1$,

$$\Pr\left[\|\mathbf{w}_i - \mathbf{w}_j\|_0 \leq (1 - \delta)d/2\right] \leq e^{-\delta^2 d/4}.$$

There are $\binom{m}{2} < m^2$ pairs of images $(i, j)$. By a union bound and with $\delta = \sqrt{8 \log m/d}$, we get that

$$\Pr\left[\|\mathbf{w}_i - \mathbf{w}_j\|_0 > d/2 - \sqrt{2d \log m} : \text{for all } i, j \in [m], i \neq j\right] > 1 - m^2 e^{-\delta^2 d/4} > 0.$$

Thus, by the probabilistic method, there exists $\mathbf{w}_1^\star, \dots, \mathbf{w}_m^\star$ such that $\|\mathbf{w}_i^\star - \mathbf{w}_j^\star\|_0 > d/2 - \sqrt{2d \log m}$ for all $i, j \in [m]$.

It remains to show that the linear classifier $F_{\mathbf{w}_1^\star, \dots, \mathbf{w}^{\star m}}$ satisfies the condition of the theorem statement. For class $i$, consider the binary image $\mathbf{x}_i = \mathbf{w}_i^\star$. Note that for binary images $\mathbf{x} \in \{-1, 1\}^d$, we have $\mathbf{x}^\mathsf{T} \mathbf{w}_i^\star = d - 2\|\mathbf{x} - \mathbf{w}_i^\star\|_0$. Thus, $\mathbf{x}_i^\mathsf{T} \mathbf{w}_i^\star = d$ and $\arg\max_{j \neq i} \mathbf{x}_i^\mathsf{T} \mathbf{w}_j^\star < 2\sqrt{2d \log m}$, and we get $\mathbf{x}_i^\mathsf{T} \mathbf{w}_i^\star - \arg\max_{j \neq i} \mathbf{x}_i^\mathsf{T} \mathbf{w}_j^\star > d - 2\sqrt{2d \log m}$. Each pixel change reduces this difference by at most 4. Thus, for all $\mathbf{x}'$ such that $\|\mathbf{x}_i - \mathbf{x}'\|_0 \leq (d - 2\sqrt{2d \log m})/4 = d/4 - \sqrt{2d \log m}/2$, we have $\mathbf{x}'^\mathsf{T} \mathbf{w}_i^\star - \arg\max_{j \neq i} \mathbf{x}'^\mathsf{T} \mathbf{w}_j^\star > 0$ and the predicted label of $\mathbf{x}'$ is $i$. $\square$

**Theorem 2.** *There exists a 2-class linear classifier $F$ for $d$-dimensional binary images such that for both classes $i$, a uniformly random binary image $\mathbf{x}$ in that class $i$ is robust to $\sqrt{d}/8$ pixel changes in expectation, i.e. $\mathbb{E}_{\mathbf{x} \sim \mathcal{U}(i)}[\min_{\mathbf{x}' : \arg\max_j F(\mathbf{x}')_j \neq i} \|\mathbf{x} - \mathbf{x}'\|_0] \geq \sqrt{d}/8$.*

*Proof.* Consider the following linear classifier

$$F(\mathbf{x}) = \begin{cases} 0 & \text{if } \vec{1}^T \vec{x} - x_0/2 < \frac{d}{2} \\ 1 & \text{otherwise} \end{cases}.$$

Informally, this is a classifier which assigns label 0 if $\|\mathbf{x}\|_0 < d/2$ and label 1 if $\|\mathbf{x}\|_0 > d/2$. The classifier tiebreaks the $\|\mathbf{x}\|_0 = d/2$ case depending on whether or not the first position in $\mathbf{x}$ is a 1 or a 0. Notice that this classifier assigns exactly half the space the label 0, and the other half the label 1.

Consider class 0 and let $\mathcal{U}(0)$ be the uniform distribution over all $\mathbf{x}$ in class 0. We have

$$\Pr_{\mathbf{x} \in \mathcal{U}(0)}[\|\mathbf{x}\|_0 = s] = \frac{1}{2^d} \binom{d}{s}$$

when $s < d/2$ and $\Pr_{\mathbf{x} \in \mathcal{U}(0)}[\|\mathbf{x}\|_0 = s] = \frac{1}{2^{d+1}} \binom{d}{s}$ when $s = d/2$. The binomial coefficient $\binom{d}{s}$ is maximized when $s = d/2$. For all $d \in \mathbb{Z}^+$, Stirling's approximation gives lower and upper bounds of $\sqrt{2\pi} d^{d+\frac{1}{2}} e^{-d} \leq d! \leq d^{d+\frac{1}{2}} e^{-d+1}$. Since $d$ is even, we get

$$\binom{d}{d/2} = \frac{d!}{(\frac{d}{2}!)^2} \leq \frac{e 2^d}{\pi \sqrt{d}}.$$

Therefore, we have that for all $s$,

$$\Pr_{\mathbf{x} \in \mathcal{U}(0)}[\|\mathbf{x}\|_0 = s] \leq \frac{1}{2^d} \binom{d}{d/2} \leq \frac{e 2^d}{\pi \sqrt{d}},$$

which implies

$$\Pr_{\mathbf{x} \in \mathcal{U}(0)} \left[ |\|\mathbf{x}\|_0 - d/2| \geq \frac{\pi\sqrt{d}}{4e} \right] \geq 1 - \frac{2\pi\sqrt{d}}{4e} \cdot \frac{e}{\pi\sqrt{d}} = \frac{1}{2}.$$

The same argument follows similarly for members of class 1. Therefore, for either class, at least half of the images $\vec{x}$ of that class are such that $|\|\mathbf{x}\|_0 - d/2| \geq \frac{\pi\sqrt{d}}{4e} \geq \frac{\sqrt{d}}{4}$. These images require at least $\frac{\sqrt{d}}{4}$ pixel flips in order to change the predicted label according to $F$, and we obtain the bound in the theorem statement. □

# B ADDITIONAL DESCRIPTION OF DATASETS AND MODELS

## B.1 DIGIT AND CHARACTER RECOGNITION SYSTEMS

**The datasets.** We preprocess the EMNIST letter dataset to only include lowercase letters, since an uppercase letter which is misclassified as the corresponding lowercase letter does not change the semantic meaning of the overall word. We randomly select 10 correctly-classified samples from each class in MNIST and EMNIST lowercase letters to form two datasets to attack.

**Models.** We consider the following five models, trained in the same manner for the MNIST and EMNIST datasets. For each model, we also list their Top-1 accuracies on MNIST and EMNIST.

- **LogReg:** We create a logistic regression model by flattening the input and follow this with a fully connected layer with softmax activation. (MNIST: 91.87% / EMNIST: 80.87%)

- **MLP2:** We create a 2-layer MLP by flattening the input, followed by two sets of fully connected layers of size 512 with ReLU activation and dropout rate 0.2. We then add a fully connected layer with softmax activation. (MNIST: 98.01% / EMNIST: 93.46%)

- **CNN:** We use two convolutional layers of 32 and 64 filters of size $3 \times 3$, each with ReLU activation. The latter layer is followed by a $2 \times 2$ Max Pooling layer with dropout rate 0.25. (MNIST: 99.02% / EMNIST: 95.04%)
  This output is flattened and followed by a fully connected layer of size 128 with ReLU activation and dropout rate 0.5. We then add a fully connected layer with softmax activation.

- **LeNet 5:** We use the same architecture as in (LeCun et al., 1998). (MNIST: 99.01% / EMNIST: 94.33%)

- **SVM:** We use the `sklearn` implementation with default parameters. (MNIST: 94.11% / EMNIST: 87.53%)

Except for the SVM, we train each model for 50 epochs with batch size 128, using the Adam optimizer with a learning rate of $10^{-3}$. The experimental results for CNN on MNIST and LeNet5 on EMNIST are shown in Section 5.

## B.2 LSTM ON HANDWRITEN NUMBERS

**The dataset.** We train an OCR model on the ORAND-CAR-A dataset, part of the HDSRC 2014 competition on handwritten strings (Diem et al., 2014). This dataset consists of 5793 images from real bank checks taken from a Uruguayan bank. The characters in these images consist of numeric characters (0-9) and each image contains between 2 and 8 characters. These images also contain some background noise due to the real nature of the images. We observe the train/test split given in the initial competition, meaning that we train our model on 2009 images and attack only a randomly selected subset from the test set (another 3784 images). The images as presented in the competition were colored, but we binarize them in a similar preprocessing step as done for MNIST/EMNIST datasets.

**The LSTM model.** We implement the OCR model described in (Mor & Wolf, 2018), which consists of a convolutional layer, followed by a 3-layer deep bidirectional LSTM, and optimizes for CTC loss. CTC decoding was done using a beam search of width 100. The model was trained with the Adam optimizer using a learning rate of $10^{-4}$, and was trained for 50 epochs. The trained model

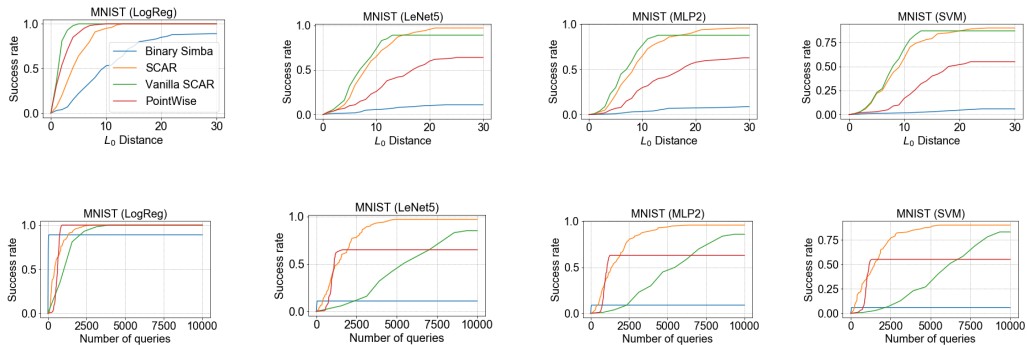

Figure 4: Success rate by $L_0$ distance and by number of queries for four different models on MNIST.

achieves a precision score of .857 on the test set of ORAND-CAR-A, which would have achieved first place in that competition.

### B.3 TESSERACT ON PRINTED WORDS

**The model.**   We use Tesseract version 4.1.1 trained for the English language. Tesseract 4 is based on an LSTM model (see (Song & Shmatikov, 2018) for a detailed description of the architecture of Tesseract's model).

**The dataset.**   We attack images of a single printed English word. Tesseract supports a large number of languages, and we use the version of Tesseract trained for the English language. We picked words of length four in the English dictionary. We then rendered these words in black over a white background using the Arial font in size 15. We added 10 white pixels for padding on each side of the word. The accuracy rate over 1000 such images of English words of length four chosen at random is 0.965 and the average confidence among words correctly classified is 0.906. Among the words correctly classified by Tesseract, we selected 100 at random to attack.

For some attacked images with a lot of noise, Tesseract does not recognize any word and rejects the input. Since the goal of these attacks is to misclassify images as words with a different meaning, we only consider an attack to be successful if the adversarial image produced is classified as a word in the English dictionary. For example, consider an attacked image of the word "one". If Tesseract does not recognize any word in this image, or recognizes "oe" or ":one", we do not count this image as a successful attack.

We restricted the attacks to pixels that were at distance at most three of the box around the word. Since our algorithm only considers boundary pixels, this restriction avoids giving an unfair advantage to our algorithm in terms of total number of queries. In some cases, especially images with a lot of noise, Tesseract does not recognize any word and rejects the input. Since the goal of these attacks is to misclassify images as words with a different meaning than the true word, we only consider an attack to be successful if the adversarial image produced is classified as a word in the English dictionary. For example, consider an image with the text "one". If Tesseract does not recognize any word in this image, or recognizes "oe" or ":one", we do not count this image as a successful attack.

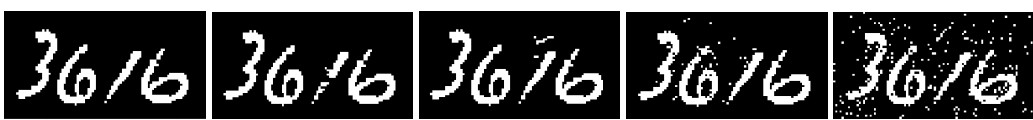

Figure 6: Examples of attacks on the LSTM for handwritten numbers. The images correspond to, from left to right, the original image, the outputs of SCAR, VANILLA-SCAR, POINTWISE, and SIMBA.

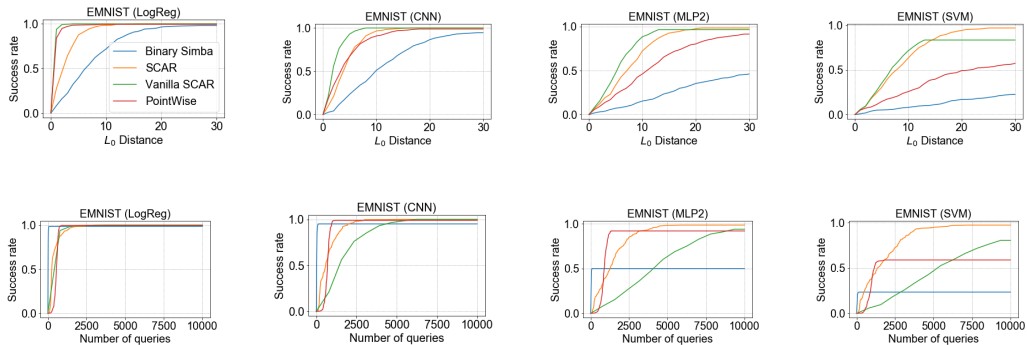

Figure 5: Success rate by $L_0$ distance and by number of queries for four different models on EMNIST.

## C  ADDITIONAL EXPERIMENTAL RESULTS

In Figure 4 and Figure 5, we provide additional experimental results on the MNIST and EMNIST datasets. In Figure 6, we give additional examples of attacks on the LSTM model for handwritten number. In Table 1, we list the 100 English words of length 4 we attacked together with the word label of the image resulting from running SCAR.

**Spatial and temporal correlations.**  In Figure 7 we plot a separate line for each pixel $p$ and the corresponding decrease in confidence from flipping that pixel at each iteration. We first note the pixels with the smallest gains at some iteration are often among the pixels with the smallest gains in the next iteration, which indicates temporal correlations. Most of the gains are negative, which implies that, surprisingly, for most pixels, flipping that pixel *increases* the confidence of the true label. Thus, randomly choosing which pixel to flip, as in SIMBA, is ineffective.

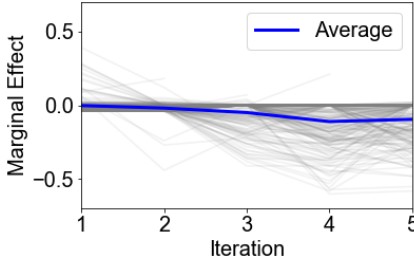

Figure 7: The gain from each pixel for the five iterations it took to successfully attack the word "idle" on Tesseract.

As discussed in Section 4, SCAR exploits spatial and temporal correlations to optimize the number of queries needed. As an example, we consider SCAR attacking Tesseract on the word "idle".

Figure 8 again shows the gain from flipping each pixel, but this time as a heatmap for the gains at the first iteration. We note that most pixels with a large gain have at least one neighboring pixel that also has a large gain. This heatmap illustrates that first querying the neighboring pixels of the previous pixel flipped is an effective technique to reduce the number of queries needed to find a high gain pixel.

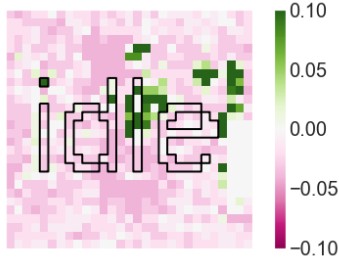

Figure 8: Heatmap of the gains from flipping a pixel on the word "idle" with Tesseract.

| Original word | Label from SCAR | | Original word | Label from SCAR | | Original word | Label from SCAR |
|---|---|---|---|---|---|---|---|
| down | dower | | race | rate | | punt | pant |
| fads | fats | | nosy | rosy | | mans | mans |
| pipe | pie | | serf | set | | cram | ram |
| soft | soft | | dare | dare | | cape | tape |
| pure | pure | | hood | hoot | | bide | hide |
| zoom | zoom | | yarn | yam | | full | fall |
| lone | tone | | gorp | gore | | lags | fags |
| fuck | fucks | | fate | ate | | dolt | dot |
| fist | fist | | mags | mays | | mods | mots |
| went | weal | | oust | bust | | game | game |
| omen | men | | rage | rage | | taco | taco |
| idle | die | | moth | math | | ecol | col |
| yeah | yeah | | woad | woad | | deaf | deaf |
| feed | feet | | aged | ed | | vary | vary |
| nuns | runs | | dray | ray | | tell | tel |
| educ | educ | | ency | ency | | avow | vow |
| gush | gust | | pres | press | | wits | wits |
| news | news | | deep | sleep | | weep | ween |
| swim | swim | | bldg | bid | | vile | vie |
| hays | nays | | warp | war | | sets | nets |
| tube | lube | | lost | lo | | smut | snout |
| lure | hare | | sqrt | sat | | mies | miles |
| romp | romp | | okay | okay | | boot | hoot |
| comp | camp | | kept | sept | | yipe | vie |
| pith | pithy | | herb | herbs | | hail | fail |
| ploy | pro | | show | how | | saga | gaga |
| toot | foot | | hick | nick | | drat | rat |
| boll | boil | | tout | foul | | limo | lino |
| elev | ale | | blur | bur | | idem | idler |
| dank | dank | | biog | dog | | twin | twins |
| gild | ail | | lain | fain | | slip | sip |
| waxy | waxy | | gens | gents | | yeti | yet |
| test | fest | | mega | mega | | loge | toge |
| pups | pups | | | | | | |

Table 1: The 100 English words of length 4 we attacked together with the word label of the image resulting from running SCAR.

Finally, in Figure 6, we show additional examples of our attacks on check processing systems.

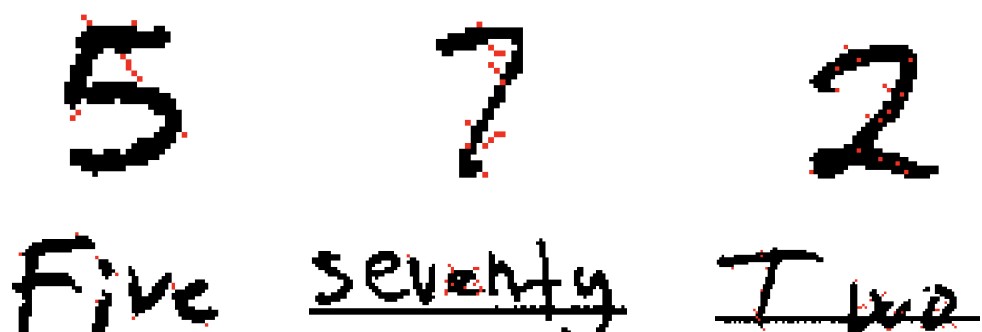

Figure 9: First digit and word of the CAR and LAR amount of checks for $562, $72, and $2 misclassified as $862, $92, and $3 by a check processing system. The pixels in red correspond to pixels whose colors differ between the original and attacked image.

