# OpenReview forum: "Adversarial Attacks on Binary Image Recognition Systems"
_ICLR.cc/2021/Conference — Reject_

### Official Review · AnonReviewer3 · 2020-10-27
**A borderline paper with clear motivation and method description but lacks rigorous evaluation**

**Rating:** 5
**Confidence:** 3

**Review:**

##########################################################################

Summary:

Adversarial attacks for binary image classification are unique from traditional attacks on color images due to its limited available space for perturbation.  This paper proposes an algorithm that efficiently searches for valid attacks (both targeted and untargeted) which cause minimum flipped pixels. The proposed method is evaluated on digit classification, letter classification, and check processing systems. The baselines compared are mostly methods that were originally developed for color images.


##########################################################################

Reasons for score:

This is a borderline paper which I tend to vote for rejection. The problem is well-defined and the method is clearly introduced. However, the evaluation of the proposed method is flawed. In particular, the classifiers being attacked seem to be simple and the perceptibility metric is questionable.


##########################################################################

Pros:

This paper proposes a novel problem with a non-trivial impact. The problem of binary image classification is also well-motivated. The authors provide a clear explanation of how attacks on the binary images are different from those on color images.

The authors provide theoretical analysis about the upper limit of pixels that need to be flipped in order to confuse the classifier.

The related work section is well-organized and the authors clearly state why previous works on color images do not apply to the binary image settings.

Both the problem formulation and description of the method are well-organized.



##########################################################################

Cons:

In the problem formulation, one constraint is that the attacks should be imperceptible to humans. This is measured by D_x(x'). However, D_x is parameterized as L0 distance which is not convincing. For instance, slightly increasing the font size of the same letter would incur a large L0 distance while making the modification imperceptible. On the other hand, transforming the letter "I" to "T" might incur a very small L0 distance while causing perceptible change. The authors should justify the use of L0 or consider experimenting with other parameterizations of D_x.

While the classifier used in section 6.3 is carefully selected and well justified, the classifiers used in section 6.2 seems too simple. In addition, instead of attacking plain classifiers, it would make the case stronger if simple defense algorithms are also considered.

The results in Section 6.5 do not have baseline algorithms to compare with. Also, it would be helpful if a reference to the check processing system is provided.

This paper lacks a conclusion.


##########################################################################

Suggestions & Questions:

In section 6.1, tau is set to 0.1. How is tau selected and how does tau impact the performance?

Please consider rewriting the last sentence of the abstract.

Please address the comments in the cons.


#########################################################################

---

> ### Author Response · Authors · 2020-11-14
> **Addressing the list of Cons**
>
> Thank you for your careful review and your positive comments --- we believe we address your reservations here and would appreciate it if you would consider improving your score in light of our response.
>
> “slightly increasing the font size of the same letter would incur a large L0 distance while making the modification imperceptible.”: We note that attacks on MNIST, for example, have been extensively studied for years and that this issue with the font size holds, to the best of our knowledge, for all the metrics (L0, L2, L_infty, …) that have been proposed. The L0, L2, and L_infty metrics are the three standard distance metrics and that, out of these three metrics, only the L0 distance is relevant in the binary setting.
>
> “the classifiers used in section 6.2 seems too simple”: We evaluate our algorithm on a wide range of classifiers, including some simple classifiers (LogReg, MLP2, SVM) and some more complex classifiers (CNN, LeNet5). As shown in the appendix B.1, the CNN and LeNet5 classifiers achieve over 99% accuracy on MNIST, which is comparable to the state-of-the-art. Finally, we also note that more complex classifiers are not necessarily more robust to attacks. In fact, in Appendix B.2, we observe that the SVM classifier is more robust to attacks than CNN and LeNet5.
>
> “it would make the case stronger if simple defense algorithms are also considered”: Evaluating simple defense algorithms is an interesting direction for future work. As a first step to understanding the vulnerabilities of binary classifiers, we think it is most relevant to evaluate classifiers without defenses, which corresponds to the classifiers used in industry such as for check processing systems.
>
> “tau is set to 0.1. How is tau selected and how does tau impact the performance?”: A larger threshold tau would produce a better attack, but at the cost of a larger number of queries. A smaller tau would improve the number of queries to the classifier, but would produce a worse attack. We pick tau such that it achieves a good balance between the two extremes. We will make this clearer.
>
> “Please consider rewriting the last sentence of the abstract.”: Thank you for the suggestion, we suggest changing it to “Consequently, it is important to raise awareness of the vulnerabilities of check processing systems and other binary image recognition systems.”

---

### Official Review · AnonReviewer2 · 2020-10-28
**Official Blind Review2**

**Rating:** 5
**Confidence:** 3

**Review:**

##################################################################
Summary:
This paper targets at a challenging task of attacking binary image classification. A score-based black-box attack algorithm SCAR is designed to fool Tesseract and US banks’ commercial check processing systems.

##################################################################
Pros:
(1) This paper introduces a challenging task of attacking binary image classifiers.
(2) The proposed method SCAR is simple yet effective, which successfully fools Tesseract and the state-of-the-art check processing systems.
(3) The paper reads smooth and is mostly well-written.

##################################################################
Cons:
(1)	The task of attacking binary image classifiers is a special case of L0 attacks, which also aim at perturbing a small number of pixels. The reviewer thinks that most L0 attack algorithms can be adapted to this binary attack task. The reviewer has concerns that since L0 attack has been widely explored, do we really need to study a much narrow task of binary attack?
(2)	The novelty of the proposed method is somewhat limited. The proposed SCAR greedily selects the modified pixels, by considering spatial and temporal correlations. However, this method is not new enough. Similar heuristic algorithms have also been developed in previous black-box attack algorithms.

---

> ### Author Response · Authors · 2020-11-14
> **Previous L0 attacks are ineffective in the binary setting**
>
> Thank you for your careful review and your positive comments --- we believe we address your main reservation that existing attacks can be employed in the binary setting. We would appreciate it if you would consider improving your score in light of our response.
>
> “attacking binary image classifiers is a special case of L0 attacks”: See paragraph titled “Previous attacks are ineffective in the binary setting” on page 2 for an explanation of why L0 attacks cannot be applied to the binary setting. If the reviewer believes that L0 attacks can be adapted to the binary setting, we would appreciate it if the reviewer would describe how this is possible.
>
> “Similar heuristic algorithms have also been developed in previous black-box attack algorithms.” If the reviewer believes that similar algorithms have been previously proposed, it would be helpful to include citations to these similar algorithms.

---

### Official Review · AnonReviewer4 · 2020-10-29
**Interesting paper, but need clarity on consequences and previous work**

**Rating:** 5
**Confidence:** 2

**Review:**

Summary: The paper describes adversarial attacks on binary (black/white) image classifiers, flipping a classifier result by changing just a few pixels. The authors go on to show that these attacks work on a real-world financial application.

Technical contributions:

* A query-optimized method for finding adversarial attacks (SCAR).
* An empirical demonstration of the method
* A proof (via a straightforward argument) that there are certain types of binary image classifiers that are provably somewhat robust to attack.

Recommendation: reject, but with weak confidence.

Reason for score: I see two main issues with this paper, both of which may be cleared up with further discussion.

* The authors make the case that this is a realistic attack on a high-value target. I believe it is worth opening a discussion on whether this raises ethical issues for publication.
* What is the relation to previous work on sparse adversarial attacks? For example, "SparseFool: a few pixels make a big difference" (Modas et al., CVPR 2019) and "One Pixel Attack for Fooling Deep Neural Networks," (Su et al., https://arxiv.org/pdf/1710.08864.pdf). While these do not relate to binary images, they seem to explore a closely related direction. The SCAR algorithm is different, but it seems important to compare with this work. The existence of other sparse methods also calls into question whether this represents a fundamentally new advance.

Areas for improvement:
* The numeric results (e.g., section 6.2) are hard to read. Putting them in a table would be helpful.
* A key goal of SCAR is "hiding the noise" (p. 4). It would be nice to have more discussion of this goal. If L_0 distance isn't the goal, why not? If there's an implicit perceptual metric at play, why not make it explicit?

---

> ### Author Response · Authors · 2020-11-14
> **Addressing two main issues**
>
> Thank you for your careful review and your positive comments --- we believe we address your two main issues here and would appreciate it if you would consider improving your score in light of our response.
>
> “whether this raises ethical issues for publication.”: The main issue is that banks and other industries that use binary image classification systems are not aware of the vulnerabilities that we demonstrate in this paper. In many areas, such as cybersecurity, academic papers have played an important role to advance the security of different systems by uncovering vulnerabilities before fraudsters and hackers start taking advantage of them.
>
> Connections to sparse adversarial attacks: Thank you for pointing out these two papers, we will include them in our discussion of related work. See the relation to our paper below:
>
> Connection to “SparseFool: a few pixels make a big difference”: SparseFool relies on an L1 relaxation of the L0 problem. L1 relaxations are not applicable in the binary setting since there are only two possible values for each pixel.
>
> Connection to “One Pixel Attack for Fooling Deep Neural Networks”: This paper focuses on attacks which modify a single pixel. In the binary setting, if it was possible to attack a classifier by modifying a single pixel, Vanilla SCAR would have found that pixel in the first iteration, since it attempts to flip every pixel. However, as we see in the experiments, in the overwhelming number of cases, Vanilla SCAR flips more than one pixel. Therefore, multiple pixel flips are necessary in order to fool binary classifiers in most cases and one pixel attacks do not apply.
>
> Numerical results are hard to read: Thank you for the suggestion, we will add a table in the next version of the manuscript.
>
> “If L_0 distance isn't the goal, why not?”: We believe that the L0 distance is a simple and fair objective for comparing the performance of SCAR to other algorithms which aim to minimize L0 distance. However, as discussed, and similarly to any distance metric we are aware of for adversarial attacks, the L0 distance is not perfect and we discuss why. We don’t believe that there exists a universal metric which perfectly measures the perceptibility of any noise added to any image.

---

### Official Review · AnonReviewer1 · 2020-10-30
**Interesting and practically important work**

**Rating:** 7
**Confidence:** 3

**Review:**

The main question this paper aims to answer is how vulnerable binary image classification systems are.  This is an important question because of the application of such binary image classifications for check processing, invoice processing, and license plate registration. One also would think that such systems are less vulnerable to adversarial attacks given the simplicity of their inputs and the fact that most adversarial attacks are based on color or grey scale images.  The authors propose an adversarial attack algorithm called SCAR that efficiently flips the binary pixels with reasonable number of queries in order to confuse the classifier to return a desirable label with high confidence. The authors show that the proposed method outperforms the existing baselines on multiple data sets. Very interestingly, they also showed that their algorithm is able to attack the online deposit systems of US bank with a high success rate. Their example of a check with the amount of $401 that is minimally modified for the amount of $701  is quite significant given that the model had to change both the word and numbers on the check.

The authors also provided some general theories on the existence of binary image classifier provably robust to any attack that modifies large, bounded number of pixels. However, it is not clear to me how this is related to the rest of paper’s discussion. The connection is missing in the paper.

---

> ### Author Response · Authors · 2020-11-14
> **Clarification about connection between provably robust classifiers and the rest of paper**
>
> Thank you for your overall positive review.
>
> “it is not clear to me how this is related to the rest of paper’s discussion”: A natural question is whether we can upper bound the number of pixel inversions needed by our SCAR algorithm to attack any classifier. The existence of provably robust classifiers show that, in general, there is no algorithm, including SCAR, that can attack any classifier by inverting a small number of pixels. We will make this connection clearer.

---

### Review · Ethics_Committee · 2021-01-06

**Decision:**

Significant concerns (Do not publish)

**Ethics Review:**

This paper was referred to the Ethics Panel because of the comment by AnonReviewer4: “The authors make the case that this is a realistic attack on a high-value target. I believe it is worth opening a discussion on whether this raises ethical issues for publication.” The primary ethical issue is whether the authors have followed procedures for responsible disclosure of risks (see https://en.wikipedia.org/wiki/Responsible_disclosure). This relates to the ICLR Code of Ethics principle of “Avoid Harm”. We have the following questions for the authors:
1.	How serious is the vulnerability? How difficult would it be for someone to exploit this vulnerability in practice? The paper claims that it would be easy to hijack the camera on an Android phone to feed it the modified image. If this is true, then disclosing this vulnerability by submitting to ICLR is already a violation of the ICLR Code of Ethics because all submissions are publicly available via OpenReview. We note that the manuscript was also released on arXiv on October 22, 2020.
2.	Are there inexpensive counter-measures to the attack? Are the confidence scores of the attacked images low so that a confidence threshold could be applied to detect the attack? It is reasonable to expect that banks have additional verification steps in the check processing pipeline. How easy do you believe it would be for banks to detect and evade these attacks?
3.	The paper does not identify the company that sells the “commercial check processing system”, so we were not able to determine whether that company has a Responsible Disclosure Policy for security vulnerabilities. Have the authors determined whether the company has such a policy?
4.	Have the authors contacted the company and informed them of the vulnerability?
5.	If so, have the authors given the company a reasonable time period to assess the impact of the vulnerability on actual banking operations?
Without knowing the answers to these questions, we cannot make a final recommendation concerning this paper. We would like to see evidence that the authors have followed procedures (formal or informal) for responsible disclosure prior to submitting the manuscript to ICLR 2021. If they have done so, there are no further ethical considerations. If they have not done so, then we recommend rejecting the paper. We realize that rejecting the paper will not prevent the harms resulting from the disclosure, because the manuscript has already been disclosed on OpenReview and arXiv. But rejecting the paper is the only mechanism available to ICLR for incentivizing authors to comply with its Code of Ethics.

Notes for ICLR: ICLR should consider whether it needs to modify its submission processes to prevent irresponsible disclosure on OpenReview. For example, authors should be asked whether the manuscript discloses any novel vulnerabilities in deployed software systems. If the authors answer affirmatively, the authors should be required to certify that they have followed procedures for responsible disclosure. Responsible disclosure practices should be addressed explicitly in the Code of Ethics.

---

### Decision · Program_Chairs · 2021-01-07
**Final Decision**

**Decision:**

Reject

**Comment:**

This paper was referred to the ICLR 2021 Ethics Review Committee based on concerns about a potential violation of the ICLR 2021 Code of Ethics (https://iclr.cc/public/CodeOfEthics) raised by reviewers. The paper was carefully reviewed by two committee members, who provided a binding decision. The decision is "Significant concerns (Do not publish)". Details are provided in the Ethics Meta Review. As a result, the paper is rejected based Ethics Review Committee's decision .

The technical review and meta reviewing process moved proceeded independently of the ethics review. The result is as follows:

This paper considers sparse (L0) attacks against binary images analysis systems, in particular OCR.  The major concern of the reviewers seems to be similarity to other methods in the literature, but reviewers did not specify any specific methods to compare to.  Because it was not possible for reviewers to address such vague concerns, and because I believe the authors did a good job differentiating their work in the rebuttal, I think the paper is of good merit.